# Metabolic interactions between dynamic bacterial subpopulations

Adam Z Rosenthal[1,2], Yutao Qi[1,2], Sahand Hormoz[1,2], Jin Park[1,2], Sophia Hsin-Jung Li[3], Michael B Elowitz[1,2,4]*

[1]Division of Biology and Biological Engineering, California Institute of Technology, Pasadena, United States; [2]Department of Applied Physics, California Institute of Technology, Pasadena, United States; [3]Department of Molecular Biology, Princeton University, Princeton, United States; [4]Howard Hughes Medical Institute, Pasadena, United States

**Abstract** Individual microbial species are known to occupy distinct metabolic niches within multi-species communities. However, it has remained largely unclear whether metabolic specialization can similarly occur within a clonal bacterial population. More specifically, it is not clear what functions such specialization could provide and how specialization could be coordinated dynamically. Here, we show that exponentially growing *Bacillus subtilis* cultures divide into distinct interacting metabolic subpopulations, including one population that produces acetate, and another population that differentially expresses metabolic genes for the production of acetoin, a pH-neutral storage molecule. These subpopulations exhibit distinct growth rates and dynamic interconversion between states. Furthermore, acetate concentration influences the relative sizes of the different subpopulations. These results show that clonal populations can use metabolic specialization to control the environment through a process of dynamic, environmentally-sensitive state-switching.
DOI: https://doi.org/10.7554/eLife.33099.001

*For correspondence:
melowitz@caltech.edu

Competing interests: The authors declare that no competing interests exist.

## Introduction

Co-utilization of carbon sources was described alongside diauxie by Jacques Monod in his PhD thesis (*Monod, 1958*), and is common in many organisms (*Peyraud et al., 2012*). In the Gram-positive bacterium *Bacillus subtilis*, two preferred carbon sources are co-utilized: glucose and malate (*Kleijn et al., 2010*). When both of these carbon sources are available they are consumed simultaneously, generating growth rates that surpass those achieved with either substrate alone (*Kleijn et al., 2010*). Under conditions of rapid growth, co-consumption of glucose and malate leads to the accumulation of high levels of acetate (Kleijn et al., 2010). As a weak organic acid, acetate can be harmful to cells even in buffered medium (*Rosenthal et al., 2008*). Acetate and related short-chain fatty acids enter the cell passively in the neutral form and then dissociate intracellularly, releasing a proton and transiently acidifying the cytoplasm (*Russell and Diez-Gonzalez, 1997*; *Roe et al., 1998*). The intracellular dissociation of acetate also disrupts the cellular anion balance, with negative effects on metabolism (*Roe et al., 1998*; *Roe et al., 2002*) and transcription (*Rosenthal et al., 2008*). When extracellular acetate levels rise to toxic levels the growing *Bacillus subtilis* culture consumes the acetate and produces acetoin, a non-toxic pH-neutral overflow metabolite that can be used as a carbon source in later growth stages (*Speck and Freese, 1973*) (*Figure 1A*).

A biphasic growth strategy, in which acetate is produced to a toxic level and then reabsorbed and replaced by a non-toxic metabolite (*Wolfe, 2005*), is common to many bacterial species and is important both for understanding the basic biology of bacterial growth in culture, and for applications in metabolic engineering (*Papagianni, 2012*). However, it has generally been studied only at the population level, implicitly assuming a homogeneous progression of the entire culture from

**eLife digest** The chemical reactions that occur within a living organism are collectively referred to as its metabolism. Many metabolic reactions produce byproducts that will poison the cells if they are not dealt with: fermenting bacteria, for example, release harmful organic acids and alcohols. How the bacteria respond to these toxins has been most studied at the level of entire microbial populations, meaning the activities of individual cells are effectively "averaged" together. Yet, even two bacteria with the same genes and living in the same environment can behave in different ways. This raises the question: do bacterial populations specialize into distinct subpopulations that play distinct roles when dealing with metabolic products, or do all cells in the community act in unison?

Rosenthal et al. set out to answer this question for a community of *Bacillus subtilis*, a bacterium that is commonly studied in the laboratory and used for the industrial production of enzymes. The analysis focused on genes involved in fundamental metabolic processes, known as the TCA cycle, which the bacteria use to generate energy and build biomass. The experiments revealed that, even when all the cells are genetically identical, different *Bacillus subtilis* cells do indeed specialize into metabolic subpopulations with distinct growth rates.

Time-lapse movies of bacteria that made fluorescent markers of different colors whenever certain metabolic genes became active showed cells switching different colors on and off, indicating that they switch between metabolic subpopulations. Further biochemical studies and measures of gene activity revealed that the different subpopulations produce and release distinct metabolic products, including toxic byproducts. Notably, the release of these metabolites by one subpopulation appeared to activate other subpopulations within the community.

This example of cells specializing into unique interacting metabolic subpopulations provides insight into several fundamental issues in microbiology and beyond. It is relevant to evolutionary biologists, since the fact that fractions of the population can switch in and out of a metabolic state, instead of evolving into several inflexible specialists, may provide an evolutionary advantage in fluctuating natural environments by reducing the risk of extinction. It also has implications for industrial fermentation processes and metabolic engineering, and may help biotechnologists design more efficient ways to harness bacterial metabolism to produce useful products.

DOI: https://doi.org/10.7554/eLife.33099.002

acetate producing to acetate detoxifying states. By contrast, single cell approaches suggest that bacterial populations can exhibit enormous heterogeneity in functional and gene expression states (*Eldar et al., 2009*; *Locke et al., 2011*; *Süel et al., 2006*; *Levine et al., 2012*; *Davidson and Surette, 2008*; *Dubnau and Losick, 2006*; *Gefen and Balaban, 2009*). This prompts the questions of whether microbial cells differentiate into metabolically distinct subpopulations, and more specifically, whether acetate production and detoxification might occur in distinct cells specializing in acetate production or detoxification, respectively.

## Results

To address these questions we constructed a library of strains with reporters for key genes involved in central carbon metabolism, acetate production, and organic acid detoxification (*Figure 1A*). We introduced a fluorescent protein (YFP) under the control of promoters for 13 different metabolic genes and stably incorporated them into the commonly used *sacA* site within the genome (*Supplementary file 1*), (*Eldar et al., 2009*; *Locke et al., 2011*). We chose to use fluorescent promoter reporters because they allow acquisition of dynamic measurements from individual living cells, are easy to construct and integrate into the *B. subtilis* genome, allow for analysis of multiple genes within the same cell, and can be used in fluorescence cell sorting for RNAseq experiments. Using quantitative single-cell fluorescence microscopy, we analyzed the distribution of expression levels of these 13 metabolic genes in individual cells at different times along the growth curve in buffered culture medium containing 22 mM glucose and 50 mM malate. To eliminate oxygen gradients, 10 mL cultures were grown in 250 mL flasks with rapid shaking (250 RPM).

Four genes had expression levels that were at or near background and were not considered further (*acoA*, *gntZ*, *pycA*, *sdhC*). Most of the genes showed unimodal distributions (*Figure 1—figure*

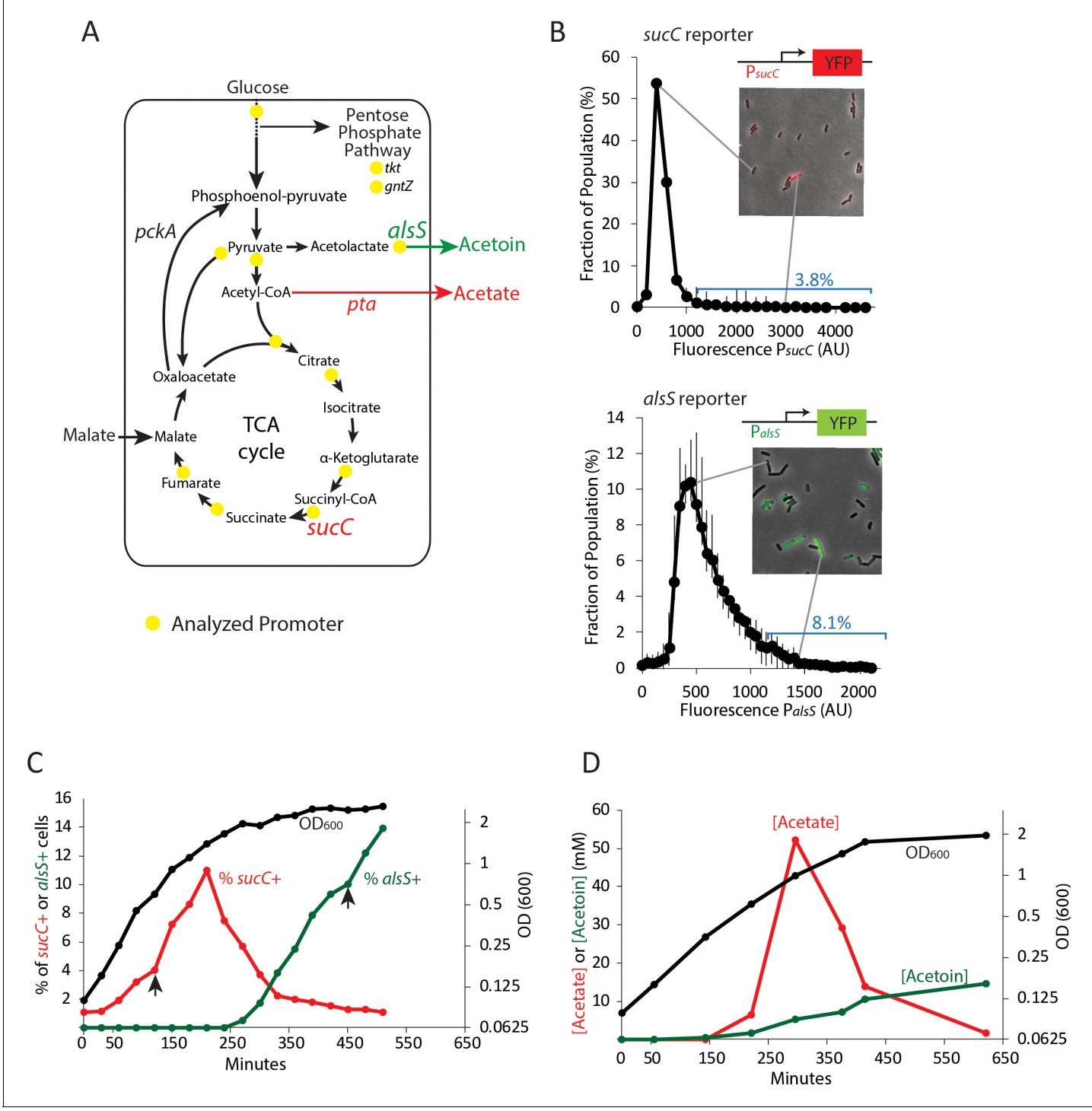

**Figure 1.** Two genes in central carbon metabolism are heterogeneously expressed in a clonal population of *B. subtilis*. (**A**) *B. subtilis* uses glucose and malate as preferred carbon sources, and under aerobic culture conditions produces acetate and acetoin as major overflow metabolites. Promoter reporter strains were made for genes participating in the reactions marked with a yellow dot (**B**) Histograms depict the heterogeneous expression of the central metabolism genes *sucC* (top panel) and *alsS* (bottom panel). Insets using merged phase and fluorescence images show typical fields of cells, including cells in the high expressing tail of the distributions. (**C**) The heterogeneous expression of *sucC* (red line) and *alsS* (green line) is maximal at different timepoints along the growth curve (black line). Black arrows denote the sampling timepoints shown in *Figure 1B*. (**D**) A line graph depicting the accumulation of extracellular acetate and acetoin in the growth media during exponential and early stationary growth (OD$_{600}$, black line). Acetate (red line) is released around mid-exponential phase, and is reabsorbed at a later time during which acetoin is produced (green line).

DOI: https://doi.org/10.7554/eLife.33099.003

*Figure 1 continued on next page*

*Figure 1 continued*

The following figure supplements are available for figure 1:

**Figure supplement 1.** Histograms of metabolic promoter reporters made for this study.

DOI: https://doi.org/10.7554/eLife.33099.004

**Figure supplement 2.** All histograms in panel B use the same X and Y values, allowing easier cross-comparison of expression levels of each reporter.

DOI: https://doi.org/10.7554/eLife.33099.005

*supplement 1*), with relatively little skew (less than ±0.7). Two genes, *sucC* and *alsS*, encoding succinate co-A ligase and acetolactate synthase, respectively, were more heterogeneous (*Figure 1B*). We observed skew values greater than 1 (4.72 and 1.14, respectively) with 3.8% of $P_{sucC}$-YFP and 8.1% of $P_{alsS}$-YFP cells exhibiting high expression levels ($\geq$2 standard deviations above the mean) at $OD_{600}$ ~0.8 (*sucC*) and $OD_{600}$ ~2 (*alsS*). In addition, for both genes, we observed cells whose expression exceeded the mean by >3 fold. While gene expression distribution can be broader immediately after gene activation than at steady-state (*Shahrezaei and Swain, 2008*), both *sucC* and *alsS* maintained heterogeneous expression for several hours after the onset of expression. For these reasons, we decided to focus on these two genes for further study.

To better understand when this heterogeneity emerges in batch culture, we performed a time course analysis of the fraction of *sucC* and *alsS* positive cells (cells $\geq$ 2 standard deviations above the mean were denoted *sucC+* and *alsS+*, *Figure 1C*). We observed that the subpopulation of *sucC+* cells only existed transiently, in mid- to late-exponential phase (*Figure 1C*), coinciding with the time and culture optical density at which acetate production was observed (when the time derivative in acetate, that is, the rate of change in acetate concentration, is positive ~150–300 min, *Figure 1D*). This observation suggested that *sucC* expression could be involved in acetate production. A parallel analysis of *alsS* expression revealed the opposite behavior, with *alsS* expression dynamics coinciding with a decrease in acetate and a concomitant increase in acetoin levels (*Figure 1C,D*). This behavior is generally consistent with the known role of *alsS* in acetoin production in response to acetate toxicity (*Speck and Freese, 1973*). Together, these results show that a dynamic change in acetate and acetoin levels in the culture overlaps with changes in the population fraction of *sucC* and *alsS* expressing cells.

A role for sucC in acetate production has not been studied previously. To understand the relationship between the subpopulation marked *sucC+* and acetate production, we used fluorescence activated cell sorting (FACS) of the $P_{sucC}$YFP reporter strain to sort cells expressing YFP from a *SucC* promoter at the time of peak acetate levels, and performed RNAseq to compare gene expression profiles (*Figure 2A*, *Figure 2—figure supplement 1*). As expected, *sucC* expression was elevated 2-fold in the *sucC+* sorted subpopulation (blue dot, *Figure 2A*). This is particularly meaningful considering the fact that the fluorescent marker used for sorting is a stable reporter, making it likely that some sorted cells may have high level of fluorescent signal even after exiting the transcriptionally active state. For most genes, we observed a broad correlation in gene expression between the two populations. However, RNAseq analysis with cuffdiff (*Trapnell et al., 2010*) and gene set enrichment analysis with GSEA (*Subramanian et al., 2005*) showed that genetic competence genes (*Berka et al., 2002*) were significantly enriched in the ~300 upregulated genes in the *sucC+* subpopulation (red dots and inset, −2A and *supplementary file 2*; GSEA p<e-16). The *sucC+* population also exhibited increased expression of the phosphate acetyltransferase gene, *pta* (green dot, *Figure 2A*), the enzyme that catalyzes the final step in overflow acetate production. Thus, *sucC* expression marks a distinct gene expression state that could be involved in acetate production.

Based on the strong correlation between *sucC* expression and competence gene expression in the RNAseq results (*Figure 2A*), we next asked whether the *sucC+* population represented the competent state. To analyze the relationship between *sucC* expression and genetic competence in single cells, we constructed four dual reporter strains, expressing CFP from the *sucC* promoter and YFP from one of four competence promoters: *comG*, *comK*, *nucA* and *rapH* (*Berka et al., 2002*; *Ogura et al., 2002*). Imaging revealed a clear positive correlation between *sucC* and the competence genes (*Figure 2B*, *Figure 2—figure supplement 2*). This positive correlation was not general to all metabolic genes, as *sucC* expression was anti-correlated with *pckA* (*Figure 2C*, *Figure 2—figure supplement 3*), a gene involved in phosphoenolpyruvate synthesis (*Meyer and Stülke, 2013*).

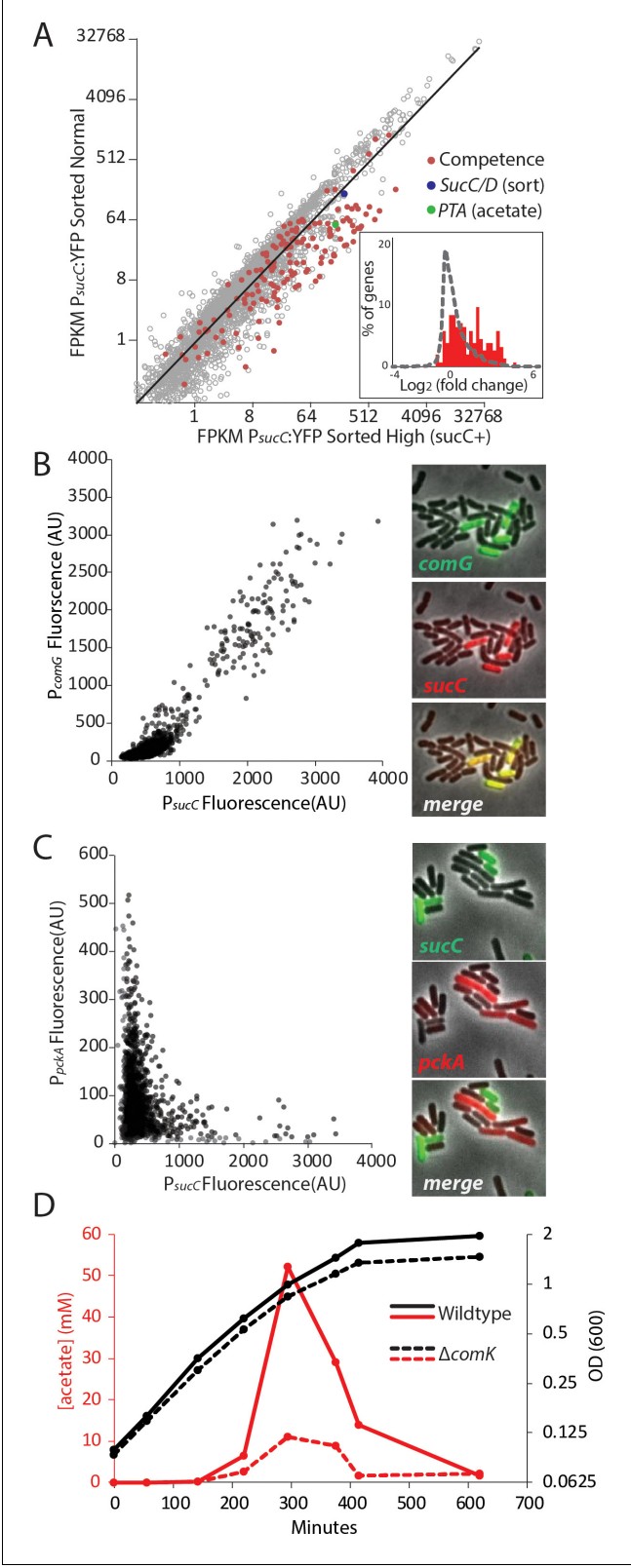

**Figure 2.** The heterogeneous expression of *sucC* is correlated with the genetic-competence regulon, and this metabolic state produces extracellular acetate. (**A**) RNAseq of cells sorted at either a high or moderate *sucC* level reveals positive correlation between sucC expression, the competence program, and the acetate metabolism gene *pta*. Cells expressing YFP under the control of the *sucC* promoter were sorted at high or normal expression

*Figure 2 continued on next page*

*Figure 2 continued*

levels. Genes for genetic competence (red) and acetate production (*pta* – green) are higher in cells expressing high levels of *sucC*. The inset histogram shows a histogram of $\log_2$ fold change for all genes (grey) and the competence program genes (red) (**B**) A scatter plot shows that the expression of *sucC* reporter is positively correlated with the *comG* reporter, a marker for the competence program. Each dot represents a single cell centered on the mean fluorescence of reporters for *sucC* and *comG*. The right panel shows fluorescent microscopy images taken from a typical field of cells (**C**) A scatter plot shows that the expression of the *sucC* reporter is negatively correlated with expression of the metabolic gene *pckA*. Each dot represents a single cell centered on the mean fluorescence of reporters for *sucC* and *pckA*. Right panel shows fluorescent microscopy images taken from a typical field of cells. (**D**) The competence gene expression program is necessary for the buildup of high levels of extracellular acetate. Growth curves demonstrate only a small difference in growth of wildtype strain (solid black line) or the competence-null Δ*comK* strain (dashed black line). However, maximal acetate buildup is approximately five fold higher in the wildtype strain (solid red line) than in a strain that is unable to produce the competent cell population (Δ*comK* dashed red lines).

DOI: https://doi.org/10.7554/eLife.33099.006

The following figure supplements are available for figure 2:

**Figure supplement 1.** FACS sorting parameters for sorting *sucC+* and normal cells for RNAseq experiments.
DOI: https://doi.org/10.7554/eLife.33099.007
**Figure supplement 2.** Competence genes are positively correlated with the TCA cycle gene *sucC*.
DOI: https://doi.org/10.7554/eLife.33099.008
**Figure supplement 3.** Competence genes are negatively correlated with the metabolic gene *pckA*.
DOI: https://doi.org/10.7554/eLife.33099.009
**Figure supplement 4.** The heterogeneous expression of *sucC*, *comG*, and *pta* are dependent on the competence master regulator comK.
DOI: https://doi.org/10.7554/eLife.33099.010

We note that *pta* (phosphate acetyltransferase gene) and *sucC* were previously observed to be up-regulated in the competent state (*Berka et al., 2002*; *Ogura et al., 2002*). Together, these results suggest that individual cells can exist in at least two distinct metabolic states, one of which represents genetically competent cells and involves increased expression of *sucC* and *pta*, among other genes.

We next assessed how competence might be linked to elevated acetate production. The competence system is controlled by a noise-excitable gene circuit that stochastically initiates transient episodes of differentiation in individual cells (*Süel et al., 2007*; *Süel et al., 2006*; *Cağatay et al., 2009*; *Maamar et al., 2007*; *Hahn et al., 1994*) To better understand the relationship between competence and acetate metabolism, we next asked whether activation of the competence system is necessary for increased *sucC* expression and acetate metabolism. Strains in which the competence master transcription factor *comK* is deleted (*Supplementary File 1*) exhibited greatly reduced acetate production (*Figure 2D*) and a loss of *sucC* as well as *comG* expression (*Figure 2—figure supplement 4*). In addition to the reduced level of acetate itself, expression of *pta*, the key step in acetate production, is greatly diminished in strains in which *comK* is deleted, both in our conditions (*Figure 2—figure supplement 4B*) and also in data from previous microarray experiments (*Ogura et al., 2002*; *Berka et al., 2002*). Although the competent state has been suggested to be involved in other functions, such as attachment, motility, antibiotic resistance, and DNA metabolism (*Redfield, 1993*; *Hahn et al., 2015*; *Bakkali, 2013*; *Finkel and Kolter, 2001*), a role in central carbon metabolism has not been reported. These results indicate that the *sucC* subpopulation is controlled by the competence system, linking competence both to an alternative metabolic state and to the control of acetate levels in culture.

To better understand the dynamics with which cells switch into the competent state and later into the *alsS+* state, we used the 'Mother Machine' microfluidic device (*Figure 3A*), to conduct long-term analysis of individual cells over tens of cell generations under chemostatic conditions (*Wang et al., 2010*; *Norman et al., 2013*). We set up the Mother Machine as described previously (*Norman et al., 2013*), but cultured cells with conditioned media obtained from batch growth of *B. subtilis* cultures at different final optical densities. Specifically, we used media from cultures at $OD_{600}$ 0.8 and $OD_{600}$ 2.0, points during the peak of *sucC+* or *alsS+* expression, respectively. This approach

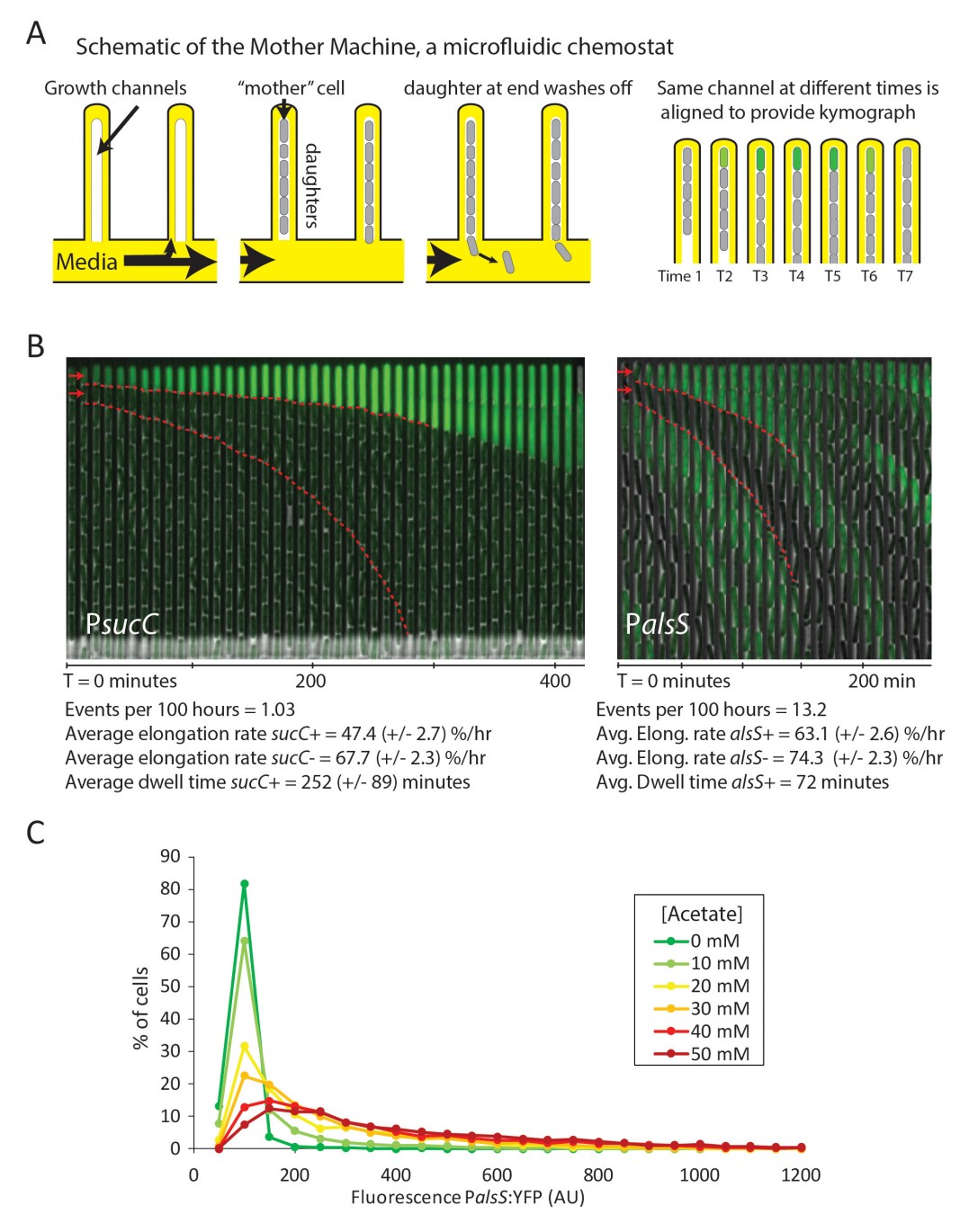

**Figure 3.** Cells switch in and out of the slower growing *sucC+* and *alsS+* states based on media conditions. (**A**) A schematic of the Mother Machine microfluidic experiment. Cells are loaded into growth channels that are capped on one end and surrounded by flowing media. A 'mother' cell settles at the capped end, and produces daughters. The daughters at the uncapped end of the growth channel are washed away by the current of media. Positions were filmed for up to 4 days, and for visualization purposes the images from each growth channel were cropped and aligned to generate a kymograph depicting time on the x-axis. (**B**) Filmstrip kymographs from representative mother-machine experiments using conditioned media at $OD_{600}$ 0.8 and *sucC* reporter strain (left panel) or conditioned media at $OD_{600}$ 2.0 using *alsS* reporter strain (right panel). Dashed red lines show the trend of growth of two daughters, a *sucC+* and *sucC-* pair on the left panel and a *alsS+* and *alsS-* pair on the right panel. As seen from the slope of the trend lines and as indicated below the kymographs, the elongation rates of both *sucC+* and *alsS+* cells are slower than their counterparts. All values reported (events per 100 hr, elongation rate, and dwell time) were averaged from at least three separate movies per condition. Representative mother machine movies are available in the supplement. (**C**) Extracellular acetate levels activate heterogeneous expression of the *alsS* promoter. A histogram shows the population of cells expressing different levels of YFP under control of the *alsS* promoter. When no acetate is added (green line), practically all cells are in the low expressing portion of the histogram. When extracellular acetate levels are added to mimic the maximal amount produced in the growth

*Figure 3 continued on next page*

*Figure 3 continued*

curve (orange and red lines), some cells remain in the low expressing portion of the histogram, but a correspondingly larger number of cells are in the long tail of high *alsS* expression.

DOI: https://doi.org/10.7554/eLife.33099.011

The following figure supplement is available for figure 3:

**Figure supplement 1.** Filmstrip kymographs from mother-machine experiments using conditioned media at OD$_{600}$ 0.8 and *sucC* reporter strain.
DOI: https://doi.org/10.7554/eLife.33099.012

provides the simplicity of long-term chemostatic analysis with the ability to compare cellular behavior at different culture time-points.

Using the Mother Machine, we analyzed cell lineages for up to 4 days (approximately 60 generations) for a total of 1,400 cell generations (*Figure 3B*). We tabulated the number of activation events per hour occurring in the terminal (top) cell of each channel, as well as the length of time that cells remained in an activated state (dwell time). Finally, we also measured the length of cells through successive frames of the movie, to obtain the mean relative elongation rates of *alsS+* and *alsS-* cells (Materials and Methods).

With *sucC*-inducing media (conditioned at OD 0.8), we observed rare episodes of *sucC* activation in some cells, lasting for approximately four hours each (252 ± 89 min, mean ±standard deviation for n = 31 events) (*Figure 3B*, left). Consistent with previous analysis of competence dynamics (*Süel et al., 2006*), *sucC+* cells divided less frequently and grew more slowly than other cells in the same movies (elongation rates of 47.4 ± 2.7 %/hr and 67.7 ± 2.3 %/hr, respectively). Cells in the activated state could switch out of the *sucC+* state and resume normal growth rates (*Figure 3—figure supplement 1*, *Video 1*). Under these conditions, we did not observe activation of *alsS* expression. By contrast, in the OD 2.0 conditioned media we

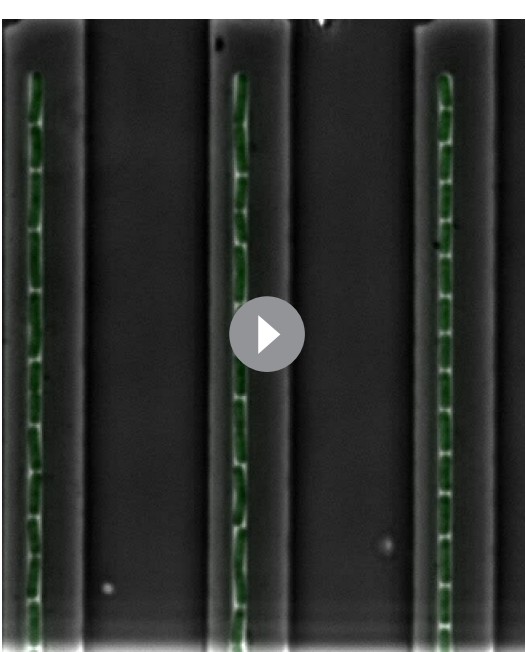

**Video 1.** A representative microscopy time-lapse video of *sucC* fluorescent reporter expression experiment demonstrates that a subset of cells switches *sucC* expression on and off. Bright green cells, expressing high levels of *sucC* (*sucC+*) have a slower growth and division rate, but can still divide.
DOI: https://doi.org/10.7554/eLife.33099.013

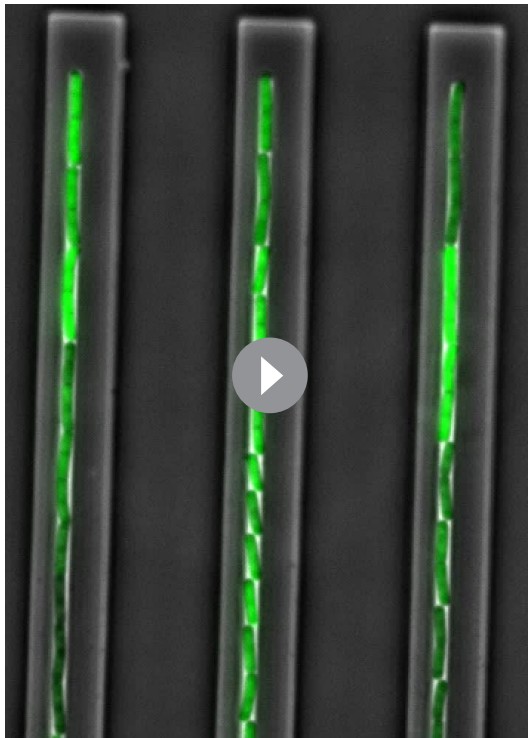

**Video 2.** A representative microscopy time-lapse video of *alsS* fluorescent reporter expression experimentdemonstrates that individual cells switch *alsS* expression on and off. Bright green cells, expressing high levels of *alsS* (*alsS+*) are able to divide.
DOI: https://doi.org/10.7554/eLife.33099.014

did not observe activation of *sucC* expression, but did observe frequent pulses of *alsS* gene expression. *alsS+* cells grew at a slightly reduced elongation rate ($63 \pm 2.6$ %/hr increase compared to $74 \pm 2.3$ %/hr for alsS- cells, *Figure 3B*, right panel, *Video 2*). Together, these results provide rates of transitions into the *sucC+* (competent) and *alsS+* gene expression states, and show that these states have altered growth rates and respond to medium composition.

These results further suggested the possibility that acetate predominantly produced by *sucC+* cells early in the growth could induce cell switching to the *alsS+* state in later growth stages, when it accumulates to toxic levels. However, many media components could differ between the $OD_{600}$ 0.8 and $OD_{600}$ 2.0 cultures. To determine whether acetate was sufficient to affect *alsS* expression, we cultured reporter cells in varying levels of acetate, in unconditioned liquid medium, and quantified the fraction of *alsS+* cells. We observed both a systematic increase in the distribution of *alsS* expression levels, and in the fraction of cells in the high expressing 'tail' of the distribution (*Figure 3C*).

The Mother Machine is ideal for analyzing cells over multiple generations in a relatively constant environment but not ideal for analyzing responses to environmental changes that happen as a consequence of growth. We therefore designed microcolony pad experiments in which acetate was added to standard microcolony medium (*Eldar et al., 2009*; *Locke et al., 2011*; *Young et al., 2011*) to 20 mM, the acetate concentration present in mid-exponential phase (*Figure 1D*) (*Speck and Freese, 1973*). In these experiments (*Figure 4*), all cells started with a low growth rate, likely owing to the initial acetate present in the growth media. As cells divided, approximately half of the population switched on high levels of *alsS* expression within 7 to 10 hr (*Figure 4*, *Figure 4—figure supplement 1*, *Videos 3–5*). As growth progressed, these *alsS+* cells exhibited a reduced growth rate, similar to that of the original culture. However, a distinct subpopulation with approximately 2.5-fold lower *alsS* expression emerged (*alsS-*), becoming greater than 70% of the population. These cells exhibited a faster division rate (*Figure 4B*, *Figure 4—figure supplement 1*) and faster elongation rate (*Figure 4C*, *Figure 4—figure supplement 2*).

The fast growing *alsS-* cells that appeared late in pad growth experiments had a large growth advantage compared to the slow growing *alsS+* cells (median elongation rates of 65 %/hr and 20 %/hr, respectively). In general, growth in the chemostatic Mother Machine using conditioned media is faster than on a pad with non-conditioned media containing acetate. However, the difference between *alsS-* and *alsS+* cells is much smaller in the microfluidic condition (74 %/hr vs 63 %/hr for *alsS-* and *alsS+*). This finding is consistent with the established role of acetoin as a molecule

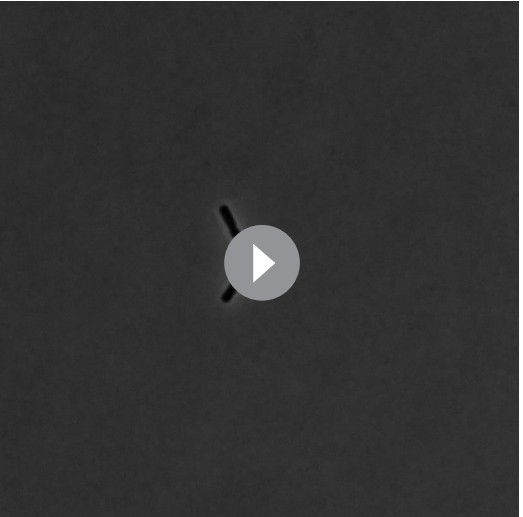

**Video 3.** One representative agarose-pad microcolony microscopy experiment of *alsS* fluorescent reporter demonstrates that individual cells switch *alsS* expression on and off. Bright green cells, expressing high levels of *alsS* (*alsS+*) divide and elongate more slowly, but are able to divide.
DOI: https://doi.org/10.7554/eLife.33099.019

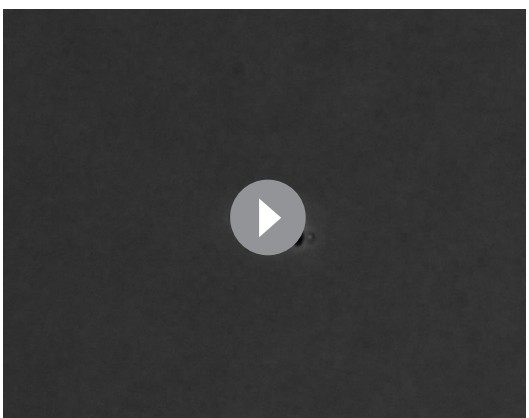

**Video 4.** A second representative agarose-pad microcolony microscopy experiment of *alsS* fluorescent reporter demonstrates that individual cells switch *alsS* expression on and off. Bright green cells, expressing high levels of *alsS* (*alsS+*) divide and elongate more slowly, but are able to divide.
DOI: https://doi.org/10.7554/eLife.33099.020

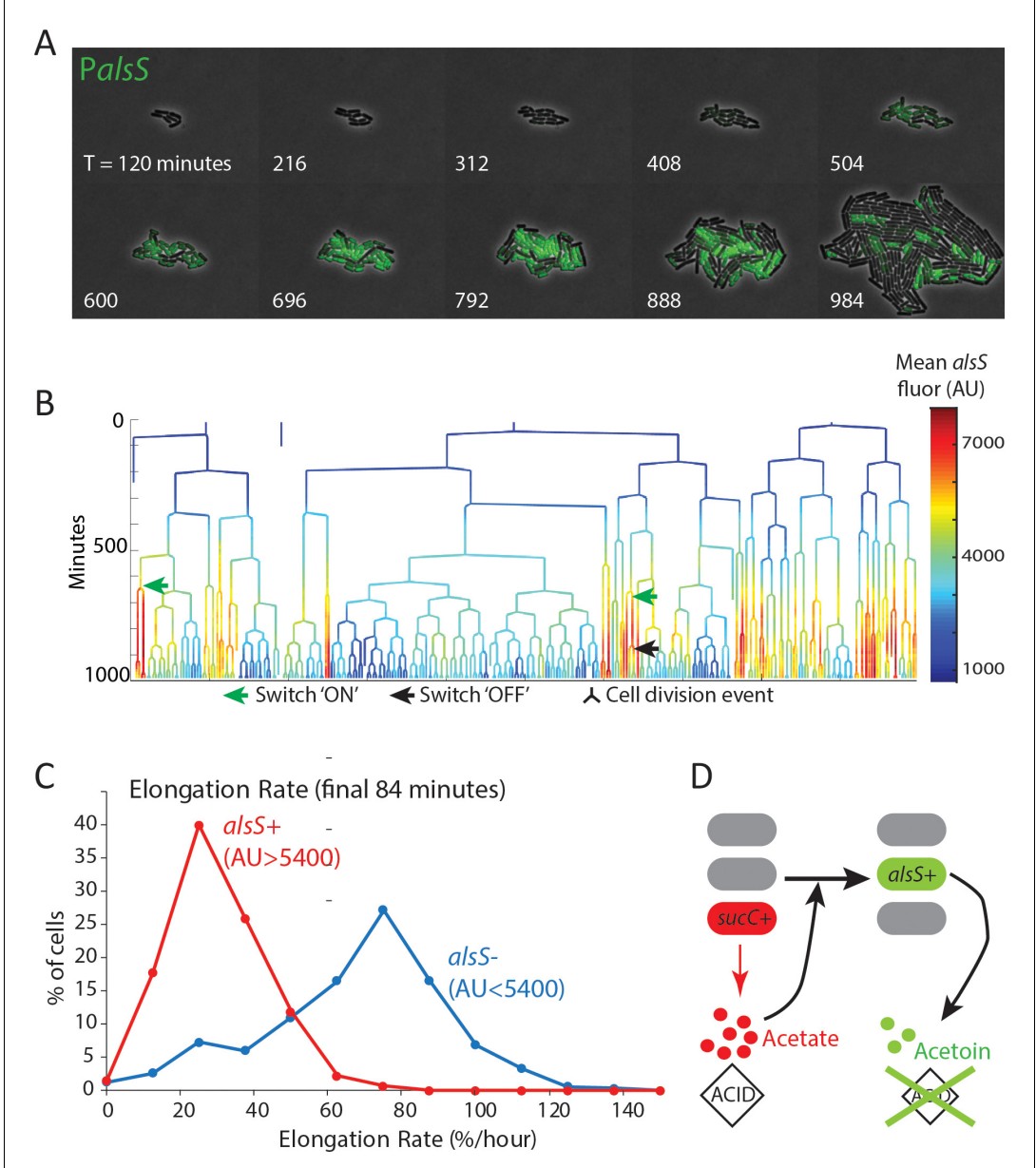

**Figure 4.** alsS +cells have slower division and elongation rate than alsS- cells. (**A**) A filmstrip of a representative timelapse experiment. Cells were grown on agarose pads containing acetate at a level that mimics mid-exponential phase. Additional filmstrips are added in supplementary movies (**B**) AlsS+ cells divide more slowly. A genealogy tree depicts cell division events in the experiment shown in panel A. alsS levels are color coded by the heatmap on the right. Cells switch in and out of high alsS expression levels. Cells expressing high alsS levels (red and orange) divide more slowly than cells with low alsS levels (blue). Similar genealogy trees are provided for three separate experiments in *Figure 4—figure supplement 1*. (**C**) AlsS- cells in the end of the experiment have faster elongation rates. Cells in the last 7 frames of the experiment which had arbitrary fluorescence levels greater than 5400 were designated alsS+ and those expressing less were designated alsS-. The elongation rate of each group of cells was determined and plotted as a histogram. alsS- cells (blue line) had a median elongation rate of 65.2 %/hr while alsS+ cells (red line) had a median elongation rate of 19.96 %/hr. Similar analysis was performed for three separate experiments. (**D**) Summary sucC and alsS interactions (schematic): In early growth stages a subset of cells become sucC+. These cells secrete acetate, which accumulates to toxic levels. High acetate levels in turn activate some cells in the population to preferentially express metabolic genes for the production of acetoin, a non-toxic pH-neutral metabolite. Finally, acetoin replaces acetate in the media.

DOI: https://doi.org/10.7554/eLife.33099.015

The following figure supplements are available for figure 4:

**Figure supplement 1.** Cells expressing high levels of alsS divide more slowly than cells expressing lower levels.

DOI: https://doi.org/10.7554/eLife.33099.016

*Figure 4 continued on next page*

*Figure 4 continued*

**Figure supplement 2.** Cells expressing high levels of alsS elongate more slowly than cells expressing lower levels in the end of the pad culture experiment.

DOI: https://doi.org/10.7554/eLife.33099.017

**Figure supplement 3.** Cells able to express *alsS* enhance the growth of neighboring cells unable to express *alsS* by improving the extracellular growth environment.

DOI: https://doi.org/10.7554/eLife.33099.018

secreted to counter the pH and anion producing toxic effect of secreted short chain fatty acids, including acetate (*Xiao and Xu, 2007*; *Speck and Freese, 1973*). In the pad environment, transient activation of genes for the production of the pH-protective acetoin in the *alsS+* cells has the potential to produce a milder growth environment which may enable other cells to grow faster (*Figure 4D*). By contrast, in the mother-machine experiments (*Figure 3*), while cells switch in and out of the *alsS* metabolic state, the chemostatic nature of the device minimizes their ability to impact the growth rates of their neighbors.

To test the idea that *alsS+* cells can enhance the growth of *alsS-* cells by influencing the extracellular environment, we seeded petri plates with equal numbers of cells from either wild-type or Δ*alsS* *B. subtilis* strains. The wildtype strain produced larger colonies than the deletion strain (*Figure 4— figure supplement 3*). However, Δ*alsS* colonies plated next to wildtype colonies exhibited a distinct growth advantage compared to the same colonies plated further away from the wildtype strain (*Figure 4—figure supplement 3*), indicating that *alsS+* cells (in the wildtype colony) alter the growth environment to benefit Δ*alsS* cells. This may occur through detoxification by *alsS+ cells* (consistent with the role of alsS in the literature). However, our data do not rule out alternative scenarios in which *alsS+* cells provide this benefit through a distinct mechanism, for example by the secretion of a shared metabolite or co-factor.

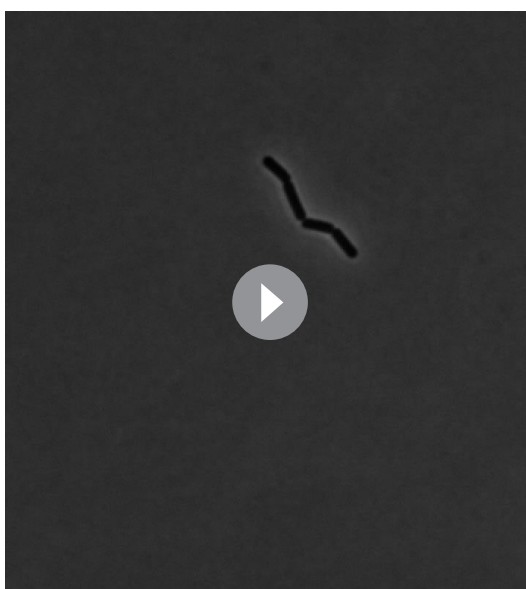

**Video 5.** A third representative agarose-pad microcolony microscopy experiment of *alsS* fluorescent reporter demonstrates that individual cells switch *alsS* expression on and off. Bright green cells, expressing high levels of *alsS* (*alsS+*) divide and elongate more slowly, but are able to divide.

DOI: https://doi.org/10.7554/eLife.33099.021

## Discussion

The natural metabolic niches in soil occupied by *B. subtilis* are not well understood, and likely heterogeneous. In general, growth of soil organisms is typically highest near deposits of organic materials which serve as carbon and nitrogen sources, including deposits of rotting fruits and vegetation. In many fruits and vegetables both organic acids (including malic acid) and sugars are very high, providing a natural setting where *B. subtilis* has access to its two preferred carbon sources simultaneously. *B subtilis* is commonly enriched (almost to levels of purity) in traditional fermentation of certain foods, including fermented soybean foods (eg natto) that are common in Asia (*Kubo et al., 2011*). In these beans sugar is present, and malic acid is also present in the root system of legumes and in soybean exudate, where it plays a role in iron homeostasis as a metal chelating agent (*Tiffin and Brown, 1962*).

The 'acetate switch' refers to the accumulation and reabsorption of acetate. It is a classic hallmark of bacterial growth in aerobic conditions, common to many bacteria including *B. subtilis* and *E. coli* (*Wolfe, 2005*). It allows for rapid initial growth until acetate levels and associated acidity reach toxic levels, at which point acetate is reabsorbed and replaced with pH-neutral overflow

metabolites such as acetoin (*Wolfe, 2005*; *Speck and Freese, 1973*). Growth strategies in which a preferred toxic overflow metabolite is produced under aerobic conditions are also used by other organisms that expel and control different fermented toxic overflow metabolites, including ethanol fermentation by yeast (*Otterstedt et al., 2004*) and lactic acid in lactobacillus species (*Borch and Molin, 1989*). Interestingly, in the fermentation of ethanol by the budding yeast *Saccharomyces cerevisiae*, ethanol is produced in dynamic bursts in which some cells switch in and out of fermentative metabolism. These bursts can be synchronized in chemostat growth (*Tu et al., 2005*), but also appear in batch culture (*Silverman et al., 2010*). In yeast, the single cell dynamics, mechanisms, and role of these bursts have not been fully elucidated. However, the presence of metabolically specialized subpopulations of cells in both eukaryotes and bacteria suggests that segregating different fermentative or respiratory pathways into individual cells may be a general strategy. It could function to avoid metabolic incompatibilities (*Brandriss and Magasanik, 1981*; *Ackermann, 2015*; *Kumar et al., 2010*), controlling cellular challenges such as reducing potential (*Liu et al., 2017*), or to optimize enzyme and substrate scaling, in which locally high concentrations of enzymes and substrates may be needed for efficient enzymatic conversion to occur (*Nikel et al., 2014*; *Ackermann, 2015*). Better understanding of the principles that govern segregation of metabolic activities will facilitate metabolic engineering and industrial fermentation approaches. This is especially true for commonly used industrial strains which naturally produce multiple fermentation products. For example, *E. coli* strains simultaneously produce five different fermentation products during mixed-acid fermentation (*Clark, 1989*).

The rapid release of toxic overflow products also has a role in the context of competition within a multi-species environment. In such environments, a quick buildup of toxic products can be advantageous to ward off competing species. In the case of human infectious disease, the buildup of byproducts such as lactic acid from normal microbiota limits infection by pathogens that are not lactic acid specialists (*O'Hanlon et al., 2013*). Likewise, in industrial fermenters and microbial food fermentation secreted overflow metabolites, including acetate and ethanol, limit contamination. Additionally, if a particular metabolic niche is transient, as in the case of acetate production in *B. subtilis* colony growth or batch culture, a strategy in which cells can switch in and out of metabolic states can be advantageous relative to an alternative scenario in which multiple strains are evolutionarily 'locked' into distinct specialist roles. This is especially true if the metabolic niche (e.g. acetate) is short-lived, because a 'locked' specialist strain would be at a disadvantage during periods of growth when acetate is absent.

In this study, we linked the presence of extracellular acetate with the activation of *alsS* in a subset of cells. In the case of competence and *sucC*, which are both controlled by the master regulator *comK*, quorum sensing plays a critical role in the activation of the competence program (*Dubnau, 1991*). Maximizing competence requires specific media conditions (*Dubnau, 1982*), raising the possibility that alongside quorum sensing peptides, secreted metabolic byproducts also play a role in this process.

Our results reveal that several metabolic processes may be affiliated with a given subpopulation. For instance, it is not clear at this point how *sucC* is related to acetate metabolism. One possible connection between *sucC* expression and acetate production might be that during co-consumption of glucose and malate, the TCA cycle may be overloaded with incoming material, causing accumulation of intermediates that can impact the glycolytic pathway, including acetate secretion. Another possibility is that *sucC* is used to regulate coenzyme-A levels at a time when a large amount of coenzyme-A is used and freed when pyruvate is converted to acetyl-coA and then acetate. Additionally, alongside the roles of the *sucC* and *pta* metabolic genes, arginine metabolism genes are also differentially regulated by competence, as seen in our results and in prior microarray experiments (*Berka et al., 2002*; *Ogura et al., 2002*), further complicating our understanding of the full physiological role of this subpopulation. Tools to systematically study metabolites in single cells would help to map the actual metabolic states of each subpopulation in a given condition and will allow us to address these questions.

Going beyond microbial systems, cell-cell heterogeneity can be advantageous as a 'bet-hedging' strategy both for microbial and cancer cells (*Veening et al., 2008*). In such cases, cell-cell heterogeneity enables the population as a whole to withstand unforeseen challenges, such as antibiotic or chemotherapeutic drugs (*Sharma et al., 2010*; *Rotem et al., 2010*), or metabolic shifts (*Solopova et al., 2014*). By contrast to simple bet-hedging, the emergence of *alsS+* acetoin

producing populations described here arises as a response to an anticipated challenge that is part of the growth progression in conditions favoring weak organic acid production. Thus, unlike in bet-hedging, metabolic state switching could provide a predictable benefit in a more deterministic dynamic environment.

## Materials and methods

### Plasmid design

Plasmids for the integration of fluorescent reporters were made as previously reported (*Eldar et al., 2009*). YFP promoter reporters were cloned into the ECE174 backbone plasmid which uses sacA integration site and encodes chloramphenicol resistance (R. Middleton, obtained from the Bacillus Genetic Stock Center). CFP promoter reporters were cloned into the pDL30 backbone which uses amyE integration sites and encodes spectinomycin resistance (obtained from the Bacillus Genetic Stock Center). A constitutive RFP reporter, using a minimal sigA promoter, was used for image segmentation as previously reported (*Locke et al., 2011*). pAZR1: (Pcggr:alsS/D) is a plasmid for the integration of a constitutive promoter (cggr promoter) to drive the constitutive expression of the alsS/D regulon from its native site. The plasmid was constructed by Gibson cloning (*Gibson et al., 2009*). A markerless deletion of alsS/D was made using the alsS/D strain of the BKE collection and the pdr244 plasmid, both obtained from the BGSC, followed by selection.

### Bacterial strains

All strains were made by genomic integration into the genome. Fluorescent reporters were integrated into either the *sacA* (YFP) or the *amyE* (CFP) loci as previously described (*Locke et al., 2011*). A constitutive RFP color was utilized, relying on constitutive expression of a partial ptrpE promoter reporter driving mCherry expression, which was inserted into the ppsB locus as previously (*Locke et al. 2011b*). Non chaining strains for microfluidic mother-machine experiments used a lytF overexpression construct as previously reported. Strain information is included as *Supplementary file 1*.

### Growth conditions

Strains were started from glycerol stocks and grown in M9 minimal media prepared according to the directions of the manufacturer (BD – difco, Franklin Lakes NJ). Base media was supplemented with 0.4% glucose (22 mM) and a cocktail of trace metals (*Leadbetter et al., 1999*) 50 mM Malate (0.66%) was added to growing cultures at OD 0.4–0.5 as per previous publications (*Buescher et al. 2012*; *Kleijn et al., 2010*) to facilitate comparison of results between our studies and other published work. Samples for fluorescence microscopy were prepared using agarose pads for either snapshot analysis (timepoint measurements) or pad movies, as previously described by our laboratory (*Young et al. 2011b*).

### Microscopy

Images were acquired using a Nikon inverted TI-E microscope via a coolsnap HQ2 camera. Commercially available software (Metamorph) controlled the stage, microscope, camera, and shutters. Fluorescent illumination was provided by a Sola Light Engine LED source (Lumencor). Temperature was kept at 37°C using an enclosed microscope chamber (Nikon) attached to a temperature sensitive heat exchanger. All experiments used a Phase 100x Plan Apo (NA 1.4) objective. Filter sets used were Chroma #41027 (mCh), Chroma #41028 (YFP), and Chroma #31044 v2 (CFP).

### Measurements of secreted acetate and acetoin

Media was collected from growing cultures by centrifuging 500 ul culture samples at 5000 g for 2 min and filtering the supernatant in 0.2 uM syringe filters. Clarified conditioned media samples were placed into glass sample vials and run at the Caltech environmental analysis center using an Aminex HPX-87H column (Bio-Rad, Rockville NY) in an Agilent 1100 HPLC with UV and Refractive Index detectors with elution using 0.013 N $H_2SO_4$ at ambient temperature and 36 ml/hour flow as described in (*Leadbetter et al., 1999*). Standards of acetate and acetoin were prepared in uninoculated growth media, and diluted to produce a standard curve.

## Microfuidic mothermachine experiments

Microfluidic experiments used the mothermachine devices described in (REFS *Wang Jun 2010*; *Norman Losick 2013*). SU80 wafers were made based on masks provided by the Losick lab. PDMS devices were prepared by pouring degassed Sylgard 184 PDMS silicone (corning corporation, Corning NY) onto wafers and curing the molds for a minimum of 8 hr at 65°C. Cured PDMS devices were bonded onto microscopy coverslips (60 × 22 mm, Gold Seal coverslips Thermo Fisher Scientific, Waltham MA) by plasma cleaning. Plasma bonding was done in a PDC32G plasma cleaner (Harrick Plasma Ithaca, NY) set to chamber pressure between 600–700 microns. Coverslips were cleaned separately for 1 min, and then the devices and coverslips were cleaned jointly for 20 s. Device bonding was immediately done by inverting the plasma treated device onto the treated coverslips. After bonding the devices were cured for an additional 4 hr at 65°C. Devices were kept for up to 2 weeks in the dark at room temperature. Before use, holes for inlet and outlet were punched using a biopsy punch and each device was passivated by loading the channels using growth-media containing 1 mg/ml BSA using a handheld micro-pippete and a 20 uM tip. Cells were loaded by flowing a concentrated cell culture (OD 2.0) and letting cells reach the growth chambers by waiting for 30 min. Devices were placed on an inverted Nikon TiE microscope and growth media was flowed using syringe pumps set to a flow rate of 50–100 ul per hour. Media used in microfluidic mother machine experiments was conditioned media taken from batch growth cultures. Media for the sucC/competence experiments contained media conditioned by growth on glucose/malate media until OD 0.8. Conditioned media used for alsS movies was from OD 2.0. Fluorescent Images were captured using a CoolSnap HQ2 and analyzed with custom MATLAB software (*Rosenthal, 2018*; copy archived at https://github.com/elifesciences-publications/Schnitzcells_2018) and in imageJ.

Events were followed for the terminal 'mother' cell position in each channel of each device, and were manually counted to provide the events per hour. Dwell time was determined by counting the number of frames from activation (passing the fluorescence threshold) and deactivation, and converted to hours and minutes by multiplying the number of frames by the frame rate of the movie. Elongation rates were measured by quantifying the length of at least 50 cells in either the activated or non-active state per each movie (at least three movies per condition), and calculated as previously determined (*Young et al., 2011*).

## RNAseq

Cultures of cells expressing YFP under the control of *sucC* (strain AZRE1) were grown in M9 glucose-malate media. Cells in mid log phase (OD 0.8) were fixed in 4% formaldehyde for 10 min at room temperature. Fixed cells were washed twice in Tris pH 7, and gently filtered using a 5.0 uM filter to remove clumps and chains. Cells were sorted on either a MoFlo astrios cell sorter or a BSfacsARIA in the USC medical school sorting facility. Cells sorted for either high YFP fluorescence or regular fluorescence (a minimum of 200,000 cells) were collected into tubes containing RNA protect (Qiagen, Hilden, Germany). Sorted samples were centrifuged and cells were rehydrated in 240 ul qiagen PKD buffer (FFPE miRNEASY kit, Qiagen). Cells were lysed by the addition of 10 ul lysosome solution for 10 min followed by bead beating for 2 min in high setting. Samples were further processed using the qiagen FFPE miRNA kit. Libraries were prepared using the Epicentre Scriptseq V2 kit, follow the directions for highly fragmented DNA. Libraries were sequenced at the Caltech Millard and Muriel Jacobs sequencing facility. Analysis followed the standard Galaxy RNAseq workflow (grooming, trimming, bowtie mapping, and cuff-diff and cuff-links) (*Afgan et al., 2016*).

## Agarose pad timelapse experiments

Agarose pad experiments were done as previously described (*Young et al., 2011*), with the following exceptions: Cells were spotted on agarose pads made with standard pad movie media (*Young et al., 2011*) which is a Spizizen's minimal media with 0.4% glucose to which acetate was added to a final concentration of 20 mM, to mimic acetate concentrations at mid exponential phase. Cells were allowed to acclimate to the agarose pad growth condition for ~2 hr, before the start of imaging. Images were acquired from multiple fields every 12 min for a total of 22 hr. Movie analysis was performed in Matlab using the Schnitzcells analysis package (*Young et al., 2011*) with slight edits. The current version of this analysis package is available at http://www.elowitz.caltech.edu

## Acknowledgements

We thank Jared Leadbetter, Ned Wingreen, Xinning Zhang, Avigdor Eldar, Joe Levine, Eric Matson, Mark Budde, Joe Markson, and members of the Elowitz lab for discussions and comments. This research was supported by the by Defense Advanced Research Projects Agency Biochronicity Grant DARPA-BAA-11–66, NIH R01GM079771, National Science Foundation grant 1547056, and a Caltech CEMI (Center for Environmental Microbial Interactions at Caltech Interactions) grant (AZR).

## Additional information

### Funding

| Funder | Grant reference number | Author |
| --- | --- | --- |
| National Institutes of Health | RO1GM079771 | Adam Z Rosenthal<br>Yutao Qi<br>Jin Park<br>Michael B Elowitz |
| Defense Advanced Research Projects Agency | Biochronicity Grant DARPA-BAA-11-66 | Adam Z Rosenthal<br>Yutao Qi<br>Jin Park<br>Sophia Hsin-Jung Li |
| Center for Environmental Microbial Interactions at Caltech | | Adam Z Rosenthal |
| National Science Foundation | 1547056 | Michael B Elowitz<br>Jin Park |

The funders had no role in study design, data collection and interpretation, or the decision to submit the work for publication.

### Author contributions

Adam Z Rosenthal, Michael B Elowitz, Conceptualization, Formal analysis, Writing—original draft, Writing—review and editing; Yutao Qi, Sahand Hormoz, Sophia Hsin-Jung Li, Formal analysis; Jin Park, Methodology

### Author ORCIDs

Adam Z Rosenthal http://orcid.org/0000-0002-6936-3665
Sophia Hsin-Jung Li http://orcid.org/0000-0001-8972-6921
Michael B Elowitz http://orcid.org/0000-0002-1221-0967

### Decision letter and Author response

Decision letter https://doi.org/10.7554/eLife.33099.027
Author response https://doi.org/10.7554/eLife.33099.028

## Additional files

### Supplementary files

• Supplementary file 1. A table listing the strains used in this work.
DOI: https://doi.org/10.7554/eLife.33099.022

• Supplementary file 2. Genes significantly differentially regulated in cells sorted by P$_{alsS}$:YFP fluorescence. Genes that are significantly differentially regulated in the sorting and RNAseq experiments are sorted based on P-values. Genes in green text have P-values of 5.0 e-5 or smaller
DOI: https://doi.org/10.7554/eLife.33099.023

• Supplementary file 3. Genes that were not significantly regulated in cells sorted by P$_{alsS}$:YFP fluorescence.
DOI: https://doi.org/10.7554/eLife.33099.024

• Transparent reporting form

DOI: https://doi.org/10.7554/eLife.33099.025

## Data availability

Data are included in supplementary files.

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
