## [Decision Letter]

Thank you for submitting your article "Metabolic Interactions Between Dynamic Bacterial Subpopulations" for consideration by *eLife*. Your article has been favorably evaluated by Naama Barkai (Senior Editor) and three reviewers, one of whom, Martin Ackerman (Reviewer #1), served as a guest Reviewing Editor. The following individual involved in review of your submission has agreed to reveal their identity: Michael G Surette (Reviewer #2). The reviewers have discussed the reviews with one another and the Reviewing Editor has drafted this decision to help you prepare a revised submission.

Summary:

This manuscript addresses a fundamental question about microbial metabolism: do individual cells in clonal populations differentiate into different phenotypes that specialize on different metabolic processes, and that complement each other in their metabolic activities? Such metabolic differentiation could have important consequences: Metabolic differences between cells in clonal populations could potentially reduce metabolic incompatibilities between different pathways, and increase the rate at which different reactions can be performed at the level of clonal populations. Such metabolic specialization has been suggested in a few earlier studies, but the evidence has often remained inconclusive. This manuscript here provides substantial evidence for metabolic differentiation in acetate metabolism in *Bacillus subtilis*, and suggests that this specialization has the consequence of detoxifying the local environment and allowing sustained growth. It can make a substantial contribution to our understanding of how cellular individuality can lead to emergent properties at the level of clonal groups.

Essential revisions:

A first question pertains to the connection between *sucC*, acetate and competence. More specifically, two steps are currently quite weak: the first is whether *sucC* activity is really associated with acetate secretion. The metabolic connection between the two is not straightforward, and may be correlative rather than causal. The authors may be able to provide better evidence from revisiting their existing data (e.g. looking at other genes, and pathway activity). One possible connection between *sucC* increase and acetate production might be the fact that (especially with the malate coming directly into the TCA cycle), the TCA cycle is jammed with incoming material, causing accumulation of intermediates that would then slow down the whole glycolytic pathway, including acetate secretion. The reviewers suggest that the authors consider this scenario, or other possible metabolic explanations of the observations (with the caveat of the delay, discussed below). Alternatively, if the authors see *sucC* primarily as a proxy for a gene expression phenotype that is putatively producing acetate, without assuming that *sucC* plays a direct role in this, then they should state this explicitly. Some of the current phrasing seems to imply a direct role of *sucC* in acetate production (for example the beginning of the third paragraph in the Results section). The other weak link is the one between *sucC* and competence. It is not obvious, for example, whether the competence system master transcription factor (*comK*) controls specifically acetate secretion, as opposed to affecting metabolism globally. Again, this may be at least partially alleviated by revisiting the data, and making sure that statements in the text do not claim more than what the data actually show.

A second question is whether *alsS+* cells indeed detoxify the environment, a statement that is central to the idea that metabolic specialization controls the microenvironment, as stated in the Abstract. At the moment, the detoxifying activity is inferred from the analysis of gene expression reporters alone. To test whether *alsS+* cells detoxify the environment, the authors could either construct an *alsS* knock-out strain, or (probably simpler) analyze existing microcolonies, and ask whether *alsS-* cells that are in a colony with many *alsS+* cells grow faster than *alsS-* cells that are in a colony with few *alsS+* cells.

The reviewers feel that these two first points will require either new experiments or new analysis of existing data, in addition to revising the text. They have two other main comments that do not require new experimental work.

About the analysis of the coefficients of variation (CV) in transcriptional reporters: the authors report that they focus on *sucC* and *alsS* based on the higher CVs of these promoters compared to a set of 11 other promoters. The reviewers suggest that the authors discuss this decision in more detail, for two reasons: first, the CV of a transcriptional reporter is expected to be influenced by the timing of gene expression (e.g., Shahrezaei and Swain, 2008), with the CV being transiently higher after the activation of transcription, and lower later on. It would thus be useful to have information about when in the growth curve (and relative to the onset of their expression) the CVs of these different reporters were determined. Second, is the CV alone an appropriate way to identify promoters that have a high degree of variation, given that previous work suggests that the CV often depends on the level of gene expression (i.e., that it is not invariant)? Using a 30% CV for identifying promoters with high variation doesn't seem obvious; *citB* and *tkt* in Figure 1—figure supplement 1 appear to possibly have a higher expressing sub-population. Also, to illustrate and justify the differential distributions of expression between different genes it may be best to include *alsS* and *sucC* with the other genes in Figure 1—figure supplement 1 (in addition to presenting them in Figure 1B). It would seem useful to see, in addition to this revised Figure 1—figure supplement 1, another version in which all graphs have the same x and y axes scales. Some genes will turn into very thin peaks, but it is very tricky right now to get a sense of how broad and high/low the different genes are relative to each other.

The reviewers would be interested in seeing a more thorough quantification of the time-resolved single-cell data (Figure 3 and Figure 4). First, there seems to be no information about the number of replicated experiments and sample size (some replicated experiments are shown in Figure 4—figure supplement 1 – but have these experiments been quantified, and are the data included in the histograms shown in Figure 4?). Second, the title of the legend of Figure 3 states that "Cells switch in and out of the […] states", but the data and analysis underlying this statement did not become clear (maybe this is the "events per 100 hours" and "dwell time"? –

---

## [Author Response]

We thank the editor and reviewers for appreciating the importance of this work and for their constructive criticism. We have made several major improvements: First, to address reviewer concerns, we added an RNAseq experiment comparing global gene expression profiles of cells in a constitutive competent state to cells in which the master regulator of competence (*comK*) has been deleted (Figure 2—figure supplement 4). Results of this experiment are fully consistent with our other findings. Second, we added an experiment demonstrating that the presence of *alsS* expressing cells is able to enhance the growth of non-expressing cells by altering the extracellular environment (Figure 4—figure supplement 3). Third, we addressed concerns about the display of data and experimental setup (Figure 2—figure supplement 1, and various improvements throughout the main text and Materials and methods). Fourth, we improved clarity by incorporating items according to the specific suggestions of the reviewers. Finally, we also made a comprehensive effort to add details that provide environmental and physiological context, as well as historical references (including a detailed discussion of physiological relevance of growth on multiple co-utilized carbons). Together, we believe that these changes have produced a greatly improved manuscript.

Essential revisions:

A first question pertains to the connection between sucC, acetate and competence. More specifically, two steps are currently quite weak: the first is whether sucC activity is really associated with acetate secretion. The metabolic connection between the two is not straightforward, and may be correlative rather than causal. The authors may be able to provide better evidence from revisiting their existing data (e.g. looking at other genes, and pathway activity). One possible connection between sucC increase and acetate production might be the fact that (especially with the malate coming directly into the TCA cycle), the TCA cycle is jammed with incoming material, causing accumulation of intermediates that would then slow down the whole glycolytic pathway, including acetate secretion. The reviewers suggest that the authors consider this scenario, or other possible metabolic explanations of the observations (with the caveat of the delay, discussed below). Alternatively, if the authors see sucC primarily as a proxy for a gene expression phenotype that is putatively producing acetate, without assuming that sucC plays a direct role in this, then they should state this explicitly. Some of the current phrasing seems to imply a direct role of sucC in acetate production (for example the beginning of the third paragraph in the Results section).

The reviewers correctly point out that the connection between *sucC* expression and acetate secretion is correlative rather than causal, with *sucC* indeed representing a proxy or marker for the competent state. We show that competent cells, which also express *sucC*, secrete more acetate (Figure 2D), but we have not shown that *sucC* is necessary or sufficient for acetate secretion. The key point is that cells in the competent state secrete more acetate. These issues were confusing as originally presented, and we have therefore clarified the presentation in the revised version, and also added a discussion of the alternative scenarios and caveats suggested by the reviewers and additional scenario, in the second to last paragraph of the Discussion.

The other weak link is the one between sucC and competence. It is not obvious, for example, whether the competence system master transcription factor (comK) controls specifically acetate secretion, as opposed to affecting metabolism globally. Again, this may be at least partially alleviated by revisiting the data, and making sure that statements in the text do not claim more than what the data actually show.

There are two issues here: First, to what extent does *sucC* expression depend on competence? Second, could the metabolic differences between competent and non-competent cells extend beyond *sucC* and acetate secretion? To address the first issue, we now better highlight Figure 2—figure supplement 4 and also add a new experiment. The original version of Figure 2—figure supplement 4 showed that deletion of the competence master regulator *comK* eliminates elevated *sucC* expression. In addition, we now also compare global gene expression between a ∆*comK* strain and a strain in which we have induced ectopic *comK* expression. This experiment (now Figure 2—figure supplement 4) shows that competence-associated genes, including *sucC* and *pta* are elevated in induced competent cells compared to ∆*comK* cells, showing that *comK* is sufficient for elevated expression of *sucC* and *pta*. We also note that these results are consistent with previous microarray analyses of competence (Berka et al., 2002 and Ogura et al., 2002), which are now discussed and cited in the text (Results).

Regarding the second issue, we agree that our results do not rule out the possibility of additional metabolic differences between competent and non-competent cells. Indeed, in our data and previous work, expression of Arginine-metabolism genes is up-regulated in competent cells. We now mention and discuss this issue in the Discussion (second to last paragraph of Discussion).

A second question is whether alsS+ cells indeed detoxify the environment, a statement that is central to the idea that metabolic specialization controls the microenvironment, as stated in the Abstract. At the moment, the detoxifying activity is inferred from the analysis of gene expression reporters alone. To test whether alsS+ cells detoxify the environment, the authors could either construct an alsS knock-out strain, or (probably simpler) analyze existing microcolonies, and ask whether alsS- cells that are in a colony with many alsS+ cells grow faster than alsS- cells that are in a colony with few alsS+ cells.

We agree that this is an important point. To address it, we performed a new experiment in which we analyze the ability of WT colonies to rescue growth defects in adjacent ∆*alsS* colonies. This result (now added as Figure 4—figure supplement 3) is discussed in the last paragraph of the Results section. When grown on agarose plates, colonies lacking *alsS* form smaller colonies compared to wildtype colonies that include the *alsS* expressing cell population. This defect can be partially rescued when the *alsS* deletion colony is grown close to a colony that contains *alsS* positive cells (a wildtype colony). Thus, we now demonstrate that *alsS+* cells can enhance the growth environment for non *alsS* expressing cells. Finally, we note in the text that while the detoxification explanation is the most plausible, we cannot rule out the possibility that *alsS+* cells provide benefits in other ways, e.g. by producing growth-promoting substances rather than removing growth-inhibitory toxins. We also further emphasize this limitation in our discussion of this data.

The reviewers feel that these two first points will require either new experiments or new analysis of existing data, in addition to revising the text. They have two other main comments that do not require new experimental work.About the analysis of the coefficients of variation (CV) in transcriptional reporters: the authors report that they focus on sucC and alsS based on the higher CVs of these promoters compared to a set of 11 other promoters. The reviewers suggest that the authors discuss this decision in more detail, for two reasons: first, the CV of a transcriptional reporter is expected to be influenced by the timing of gene expression (e.g., Shahrezaei and Swain, 2008), with the CV being transiently higher after the activation of transcription, and lower later on. It would thus be useful to have information about when in the growth curve (and relative to the onset of their expression) the CVs of these different reporters were determined. Second, is the CV alone an appropriate way to identify promoters that have a high degree of variation, given that previous work suggests that the CV often depends on the level of gene expression (i.e., that it is not invariant)? Using a 30% CV for identifying promoters with high variation doesn't seem obvious; citB and tkt in Figure 1—figure supplement 1 appear to possibly have a higher expressing sub-population.

We now include a discussion of our choice to use stable promoter-reporters and to focus on *alsS* and *sucC* as well as the caveat raised by the referees regarding timing and gene expression variability (including the reference suggested by the referees) in the second paragraph of the Results section. We agree that CVs may not be the best measure to use, and now report in addition the skew of the distributions, which is more indicative of the long-tail property that we are focused on. We also agree that *citB* also seems to be potentially interesting, so we now explain that we focused on the two most variable genes for this study, with the hope to follow up on *citB* in a subsequent study. Finally, we more clearly include all the information regarding to time on the growth curve for each reporter.

Also, to illustrate and justify the differential distributions of expression between different genes it may be best to include alsS and sucC with the other genes in Figure 1—figure supplement 1 (in addition to presenting them in Figure 1B). It would seem useful to see, in addition to this revised Figure 1—figure supplement 1, another version in which all graphs have the same x and y axes scales. Some genes will turn into very thin peaks, but it is very tricky right now to get a sense of how broad and high/low the different genes are relative to each other.

We agree and now provide two versions of this supplementary figure, showing all of the promoters including *sucC* and *alsS*, with either individually normalized X and Y axes (Figure 1—figure supplement 1), or fixed axis limits (Figure 1—figure supplement 2).

The reviewers would be interested in seeing a more thorough quantification of the time-resolved single-cell data (Figure 3 and Figure 4). First, there seems to be no information about the number of replicated experiments and sample size (some replicated experiments are shown in Figure 4—figure supplement 1 – but have these experiments been quantified, and are the data included in the histograms shown in Figure 4?). Second, the title of the legend of Figure 3 states that "Cells switch in and out of the […] states", but the data and analysis underlying this statement did not become clear (maybe this is the "events per 100 hours" and "dwell time"?

We expanded the discussion of the single-cell data quantification in both the Materials and methods and within the main text, where we now explain the switching rate and dwell time analysis. We also include the number of replicates used for each condition and experiment, and explicitly state that the data is from multiple experiments in both Materials and methods and in the figure legends.